# Characteristics of Chemosensory Perception in Long COVID and COVID Reinfection

**DOI:** 10.3390/jcm12103598

**Published:** 2023-05-22

**Authors:** Mikki Jaramillo, Thankam P. Thyvalikakath, George Eckert, Mythily Srinivasan

**Affiliations:** 1Department of Oral Pathology, Medicine and Radiology, Indiana University School of Dentistry, Indiana University Purdue University at Indianapolis, Indianapolis, IN 46202, USA; mikjaram@iu.edu; 2Department of Dental Public Health and Dental Informatics, Indiana University School of Dentistry, Indianapolis, IN 46202, USA; tpt@iu.edu; 3Department of Biostatistics and Health Data Science, Indiana University School of Medicine, Indianapolis, IN 46202, USA; geckert@iu.edu

**Keywords:** long COVID, reinfection, taste dysfunction, smell perception

## Abstract

Emerging data suggest an increasing prevalence of persistent symptoms in individuals affected by coronavirus disease-19 (COVID-19). The objective of this study was to determine the relative frequency of altered taste and smell in COVID reinfection (multiple COVID positive tests) and long COVID (one COVID positive test). We sent an electronic survey to patients in the Indiana University Health COVID registry with positive COVID test results, querying if they were experiencing symptoms consistent with long COVID including altered chemosensory perceptions. Among the 225 respondents, a greater long COVID burden and COVID reinfection was observed in women. Joint pain was reported as the most common symptom experienced by 18% of individuals in the long COVID cohort. In the COVID reinfection cohort >20% of individuals reported headache, joint pain, and cough. Taste perception worse than pre-COVID was reported by 29% and 42% of individuals in the long COVID and COVID reinfection cohorts, respectively. Smell perception worse than pre-COVID was reported by 37% and 46% of individuals in long COVID and COVID reinfection cohorts, respectively. Further, Chi-square test suggested significant association between pre-COVID severity of taste/smell perception and headache in both cohorts. Our findings highlight the prevalence of persistent chemosensory dysfunction for two years and longer in long COVID and COVID reinfection.

## 1. Introduction

Globally as of April 2023, there have been 762,201,169 confirmed cases of coronavirus disease-2019 (COVID-19) caused by the severe acute respiratory syndrome coronavirus-2 (CoV-2) [1]. While mass vaccination has lessened the acute illness and most individuals with COVID return to their baseline state of health, a proportion of individuals exhibit persistent symptoms for extended period [2,3,4]. Variably referred to as post COVID, post-acute sequelae of SARS-CoV-2 infections (PASC) or long COVID, the condition is defined as symptoms that occur in individuals with a history of probable or confirmed CoV-2 infection that begins within three months of the onset of COVID and lasts at least 2 months and cannot be explained by an alternate diagnosis [5]. Amongst the myriad of symptoms that have been reported in confirmed and suspected cases of long COVID, the most frequent are fatigue, cognitive dysfunction, depression, chemosensory dysfunction, shortness of breath and cough [5,6,7]. Furthermore, age, sex, pre-infection comorbidities (diabetes, asthma) and severity of acute CoV-2 infection (symptomatic/asymptomatic, hospitalization) are confounding factors that could contribute to the development and/or persistence of heterogeneous post-COVID-19 conditions [2,8,9].

Intense disturbances of taste, smell, and chemesthesis were widely reported in acute COVID-19 infection with a prevalence of over 30% in most studies [10,11,12,13]. These chemosensory dysfunctions have been shown to be persistent for several months after primary infection [3,11,14,15,16]. Further, some individuals developed qualitative taste disturbances such as phantosomia or phantosmia months after primary CoV-2 infection [17,18]. In addition, the symptoms may fluctuate and relapse over time. To our knowledge, few studies have evaluated the prevalence of chemosensory dysfunction in patients with repeat COVID infections. In this study, we report a detailed analysis of population-based, self-reported survey data from hospitalized and non-hospitalized individuals with a history of a CoV-2 positive test. Our objectives were to assess the frequency of altered taste and smell in individuals with single and repeat CoV-2 infections and correlate them with common post-COVID neurological symptoms. 

## 2. Materials and Methods

### 2.1. Study Design

An electronic survey was sent to individuals aged 18 years and older, who had previously agreed to be notified about COVID-19 studies registered at IU School of Medicine’s COVID-19 Research Registry. The questionnaire was designed in consultation with the Indiana Clinical and Translational Sciences Clinical Research Core and included demographics, COVID-19 test results and questions on commonly reported long COVID symptoms. Only individuals who self-identified as testing positive for COVID completed the survey. The survey was divided into two parts: the first part included 9 items on sociodemographic characteristics and the second part consisted of 30 items that measure the following eight dimensions: general health (GH: perception of overall personal health); physical activities (PA: limitations in performing everyday work and other daily activities); emotional/mental health (E/MH: MH, feeling depressed or anxious); specific general health symptoms previously reported in long COVID (GH-LC, headache, muscle pain, chest pain, joint pain, cough); social function (SF: effect on social activities); chemosensory perception (CP: effects on taste and smell sensations); and oral health (OH: oral health). For positive COVID tests, the responders reported the date of initial positive testing and the number of times they tested positive subsequently. Specific questions on physical/emotional health and limitation in activities recorded binary responses as yes/no. Response to symptom specific items were recorded on a five-point scale for long COVID symptoms (all the time/always, most of the time/often, some of the time/sometimes, a little of the time/rarely and none of the time/never). Response to taste- and smell-specific questions were recorded on a four-point scale (same as before, worse than before, better than before and total loss). Study data were collected and managed using REDCap (Research Electronic Data Capture) tools hosted at Indiana clinical and translational sciences institute [19,20].

### 2.2. Statistical Analysis

Data from surveys completed between August and September 2022 were analyzed in this study. Descriptive statistics were used to express categorical data (frequency and percentage) of long COVID symptoms and changes in chemosensory perceptions. Age was summarized using means and standard deviations. Chi-square test was used to study the association between the parameters of duration, number reporting specific symptoms, and the intensity of the symptoms. Data are provided as mean ± SD or as Chi-square score and *p* value.

## 3. Results

### 3.1. Participant Characteristics

The survey was distributed to 13,561 volunteers. Two hundred and twenty-five responders self-reported the date and the number of times they had COVID-19 positive test results. Of the 225 responders, 57 were males and 157 were females, and the rest chose not to identify themselves. The mean age was 45.8 years (range: 19–84 years). One hundred and ninety-four identified as white, 18 as African American, 7 as Asian, 3 as Hispanic and rest chose not to report their race. 

One hundred and seventy-two individuals reported a single COVID-19 positive test. The duration since testing positive ranged between 6 and 906 days with a median of 222 days (Table 1A). Only data from individuals testing positive at least 60 days prior to the date of responding to the survey were selected for analysis (Table 1B). This included a total of 127 respondents and constituted the long COVID cohort. There were more females (92) than males (29) in this cohort. This is consistent with the reports of higher preponderance of females being diagnosed with long COVID [21,22]. Eight individuals reported hospitalization due to COVID in this cohort. Amongst these individuals, the duration of persistent symptoms was less than two months in two individuals, one individual each experienced taste perception worse than pre COVID for 4 months and 28 months, respectively, and four did not experience any change post COVID. We excluded all individuals with a history of hospitalization due to COVID in further analyses to minimize confounding factors.

Fifty-two individuals self-reported testing positive for COVID-19 two times or more. Of these, 78.8% (41) tested positive twice, 17.3% (9) tested positive three times, and two individuals (3.8%) reported four positive COVID-19 tests (Table 1A). The duration since the first positive COVID-19 test ranged between 21 and 875 days with a median of 556 days. Responses from five individuals with two positive COVID-19 tests and duration of symptoms less than 60 days were not included for further analysis. The remaining cohort, with 47 (11 males and 36 females) individuals with the minimum duration of 101 days or three months, constituted the COVID reinfection cohort (Table 1B). Four individuals reported a history of hospitalization due to COVID in this cohort and data from these individuals were excluded in further analysis to minimize confounding factors. 

All individuals in this study were vaccinated, with 57% in the long COVID cohort and 55% in the COVID reinfection cohort receiving two boosters. In the COVID reinfection cohort, 11% received four boosters and in the long COVID cohort, 2% received five boosters (Table 1).

### 3.2. Incidence of Long COVID Symptomatology

Common symptoms that comprise the post-COVID conditions include tiredness or fatigue that interferes with daily life activities that get worse after physical or mental effort (also known as “post-exertional malaise”), joint pain, muscle pain, respiratory symptoms such as cough and chest pain, neurological symptoms such as headache, depression, anxiety and changes in taste or smell [2,3,5,23]. In our long COVID cohort, joint pain was reported as the symptom experienced most often or always by 19% of individuals, and sometimes by 16%. The symptoms of headache, cough, and muscle pain were reported to be experienced sometimes by 16–21% and often by 7–9% of individuals (Figure 1A). Further, joint pain, muscle pain, and headache were experienced at least sometimes for longer than one year in 10% of individuals with long COVID.

In the COVID reinfection cohort the symptoms of joint pain, muscle pain, cough, and headache were reported to be experienced often or always by 23%, 23%, 21% and 17% of individuals, respectively. The frequency of symptoms that was experienced sometimes was higher for chest pain and headache (35%), followed by cough (20%), joint pain (19%), and muscle pain (19%) (Figure 1B). In this cohort, headache, joint pain, cough and muscle pain were reported to be present often and always by 8%, 11%, 22% and 15% for two years. However, it is relevant to note here that since all individuals were responding to a survey question on the experience of these symptoms post-positive COVID-19 test, it is not known whether they are reporting symptoms being experienced after the first infection that persisted and/or increased, or symptoms that began after subsequent re-infections.

### 3.3. Chemosensory Symptoms in Long COVID and COVID Reinfection

#### 3.3.1. Changes in Taste Perception

In our long COVID cohort, 29% (35/121) reported worse taste perception (Figure 2A) with duration ranging between two and thirty months. Equivalent number of individuals reported experiencing worse taste perception for >60 days and <12 months and for periods >12 months. Further, one individual experienced total loss and three reported better taste perception than pre-COVID. The Chi-square test for the groups of no change in taste perception and taste perception worse than pre COVID across the durations of 2–6 months, >6–12 months, >12–24 months and >24 months was 13.72, *p* < 0.003 (Figure 2B, Table 2A). 

In the COVID reinfection cohort, 42% (18/43) reported worse taste perception than pre COVID with duration ranging between 79 and 875 days (Figure 2C). Further, two individuals reported total loss and one reported experiencing better taste perception than pre COVID. Of the 18 individuals with taste perception worse than pre COVID, one reported being infected by SARS-CoV-2 four times, three individuals were infected thrice, and 13 were infected twice by the COVID-19 virus. The Chi-square test for the groups of no change in taste perception and taste perception worse than pre COVID across the durations of 2–6 months, >6 to 12 months, >12–24 months and >24 months was 3.2, *p* < 0.36 (Figure 2D, Table 2A). 

#### 3.3.2. Changes in Smell Perception

In the long COVID cohort, 37% (45/121) reported worse smell perception worse than pre COVID for durations between two and thirty months (Figure 3A). Further, one individual experienced total loss and three reported better smell perception than pre COVID. With respect to the frequency and duration, the Chi-square test for the groups of no change in smell sensation and perception of smell worse than pre COVID across the durations of 2–6 months, >6 to12 months, >12–24 months and >24 months was 11.1, *p* < 0.012 (Figure 3B, Table 2B). 

In the COVID reinfection cohort, 46% (19/43) reported worse smell perception than pre COVID for durations ranging between 79 days and 875 days (Figure 3C). Further, two individuals reported better smell perception than pre COVID. The Chi-square test for the groups of no change in smell and smell perception worse than pre COVID across the durations of 2–6 months, >6–12 months, >12–24 months and >24 months was 4.1, *p* < 0.25 (Figure 3D, Table 2B). 

#### 3.3.3. Correlation between Vaccination and Smell/Taste Perception in Long COVID and COVID Reinfection

In both the long COVID and the COVID reinfection cohorts, a higher percentage of individuals that received two boosters experienced worse taste (14% and 22%, respectively) or smell perception (19% and 22%, respectively) (Figure 4A,B). Interestingly, the number of individuals reporting chemosensory dysfunction decreased precipitously with increasing number of boosters, with 4% and 2% experiencing altered smell in the long COVID and COVID reinfection cohorts, respectively. However, this is attributed to the higher number of individuals (≥55%) (Table 1) receiving two boosters, as opposed to 20% receiving three or four boosters in our study cohort. The association between the number of boosters and the duration of worse taste/smell perception was not significant with Chi-square values of 12.8 and 9 respectively, *p* < 0.3.

#### 3.3.4. Correlation between Taste and Smell Perception and with Other Neurological Symptoms

In both the long COVID and COVID reinfection cohorts, a greater number of individuals reported experiencing both taste and smell perception worse than pre COVID than either taste or smell dysfunction alone as is evident by the r value of 0.7 (Figure 5A–D). The Chi-square test statistic for comparison of long COVID and COVID reinfection groups for persistent taste dysfunction post COVID across the three discrete time periods of less than one year, one to two years, and over two years was 7.5, *p* < 0.024. The Chi-square test statistic for the comparison of long COVID and COVID reinfection groups for persistent smell dysfunction across the three discrete time periods of less than one year, one to two years, and over two years was 8.7, *p* < 0.045 (Table 3).

In the long COVID cohort, the Chi-square test statistic for the relation between pre COVID severity of taste perception (no change, worse, total loss) and headache (never and rarely; sometimes and often and always) was 17.2, *p* < 0.0002. The Chi-square value for similar comparison in the COVID reinfection cohort was 9.3, *p* < 0.01. The Chi-square test statistic for the relation between pre-COVID severity of taste perception and joint pain 28.4, *p* < 0.7 × 10^−7^ and 15.5, *p* < 4.40 × 10^−4^ in the long COVID and COVID reinfection cohorts, respectively. With respect to smell perception, in the long COVID cohort, the Chi-square test statistic for relation between pre-COVID severity and headache was 18.7, *p* < 4.40 × 10^−4^. The Chi-square test value for severity of changes in smell perception and joint pain with respect to pre-COVID levels was 30.9, *p* < 2.20 × 10^−6^ and 15.5, *p* < 0.0004 in the long COVID and COVID reinfection cohorts, respectively (Table 4).

## 4. Discussion

In this single survey study, we assessed non-hospitalized individuals with long COVID (one positive COVID test) and with COVID reinfection (two or more positive COVID tests) for persistent symptoms and chemosensory dysfunction. Our data show that 29% and 42% reported worse taste and 37% and 46% reported worse smell perception than pre COVID in the long COVID and COVID reinfection cohorts, respectively. This is consistent with the previous reports of persistent dysgeusia and anosmia in long COVID studies [6,24]. 

Viral infections such as influenza have been associated with altered smell and taste [25,26,27]. Typically, in respiratory infections including the previous coronavirus infections, smell disturbances occur due to localized impediment to airflow conduction by excessive mucus and/or to the swelling of the respiratory mucosa [28]. In contrast, in a study comparing gustatory functions in patients affected by COVID-19 and/or the common cold, it was observed that the sweet and bitter taste scores were significantly worse in COVID-19 patients without nasal congestion or discharge [29]. This suggests that the taste disturbances reported by COVID patients may reflect actual impairment of gustatory abilities, rather than olfactory dysfunction. Other unique features of the altered taste and smell in the COVID-19 pandemic are their higher incidence, persistence, and occurrence of specific taste dysfunction without smell loss [11,15,30,31]. Consistently, we observed that in the long COVID cohort, while 29% reported both taste and smell dysfunction post COVID, 5% of individuals reported gustatory dysfunction alone. In the COVID reinfection cohort 8% reported persistent worse taste perception post COVID but no change in smell perception.

A recent observational study showed that repeat CoV-2 infections increases risk for cardiac, pulmonary, or neurological problems [32]. We observed that in our long COVID and COVID reinfection cohorts, at least 20% reported persistent (often/always) experience of at least two common long COVID symptoms. To our knowledge, this is the first study reporting on the altered taste and smell perception amongst individuals with COVID re-infection. We observed that while >30% experienced persistent dysgeusia, 18% reported persistent dysosmia in the long COVID cohort. However, in the COVID reinfection group, an equivalent number of individuals reported experiencing persistent dysgeusia and olfactory dysfunction for longer than one year. 

The available literature suggests that the severity of smell and taste alterations is reduced in CoV-2-infected individuals during the periods of Omicron variant dominance [33,34]. However, in our study cohort, the experience of altered (worse than pre COVID) taste and smell perception does not seem to be related to the frequency of infection or the number of boosters received. This could be attributed to the difference in the global prevalence of different variants and that the positive COVID-19 test in our cohort was not-restricted to one variant. Interestingly, Notarte et al., in a systematic review, show that there is a low level of evidence to suggest that vaccination before SARS-CoV-2 infection could reduce the risk of developing subsequent long COVID [35].

The interest in elucidating the pathogenesis of altered taste and smell has escalated since COVID-19, with several hypothesis projected including viral infection-induced changes in tongue biofilm, local inflammatory responses, and neurological disturbances [30,36]. Interestingly, we observed that in both the long COVID and COVID reinfection cohorts, the association between persistent taste/smell dysfunction and headache was significant, thus suggesting a role for neurological mechanism.

Limitations: While our study provides significant new knowledge with respect to COVID reinfection and chemosensory dysfunction, multiple limitations are recognized. 1) Our data does not specify the time period between the COVID-19 positive tests in the reinfection cohort, and hence could potentially include repeat positive tests in relation to the first infection. However, since we analyzed only data from responders reporting symptoms that lasted longer than two months, and >60% of individuals experienced symptoms for longer than one year and 10% for longer than two years, it is likely that the responses are symptoms that persisted after first positive COVID-19 test in the COVID re-infection cohort. A second limitation is that the retrospective questions about pre-COVID health status may be biased by the current health status. While we report age, sex, and race distribution, we did not perform symptom analyses by subgroup. Further, while the relative prevalence of altered chemosensation in our study is concomitant with previous studies [10,16,17], comorbidities such as diabetes that existed pre COVID or newly developed post COVID could have potentially contributed to the altered chemosensation in both long COVID and COVID reinfection cohorts [2,22]. The objectives of our study were to examine the relative frequency of altered taste and smell associated with COVID reinfection compared with that of a single infection. The results do not represent an assessment of severity of a second infection versus that of a first infection.

## 5. Conclusions

In summary, our data are consistent with the observations that the long COVID characterized by persistent multi-organ symptoms affects a proportion of COVID-19 infected individuals despite vaccination. Repeat infections are more likely associated with altered chemosensory perception for extended periods. Since chemosensory dysfunction has been strongly associated with neurological pathologies, depression, and inadequate quality of life, protective measures to prevent reinfection with SARS-CoV-2 are warranted. Future longitudinal follow-up studies or in-depth electronic health data mining studies are needed to better elucidate these relationships.

## Figures and Tables

**Figure 1 jcm-12-03598-f001:**
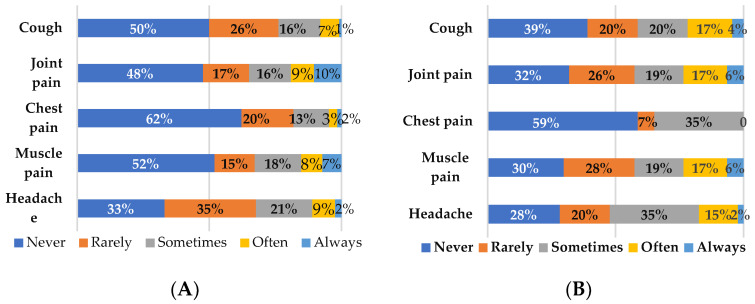
Proportion of individuals reporting the indicated symptom in the (**A**) long COVID and (**B**) COVID reinfection cohorts.

**Figure 2 jcm-12-03598-f002:**
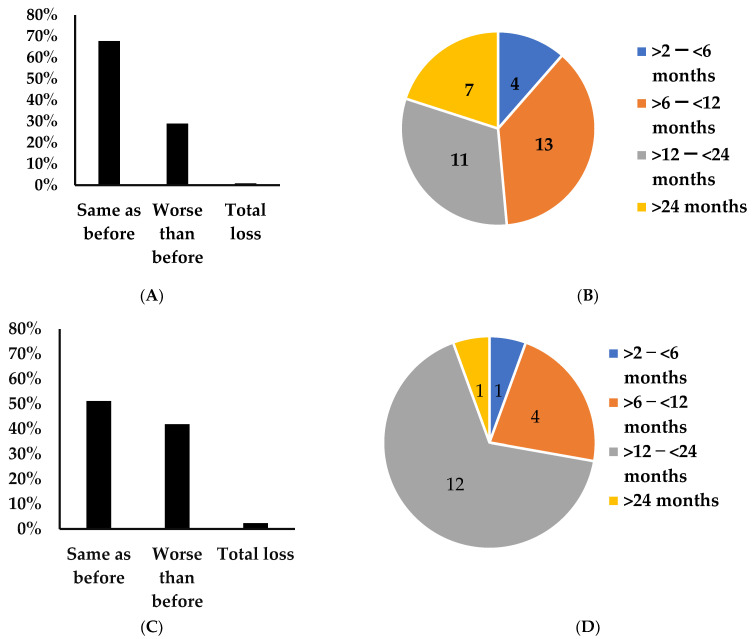
Characteristics of changes in taste perception. Distribution of altered taste among individuals in the long COVID (**A**) and COVID reinfection (**C**) cohorts. Distribution of duration of taste perception worse than pre COVID in the long COVID (**B**) and COVID reinfection (**D**) cohorts.

**Figure 3 jcm-12-03598-f003:**
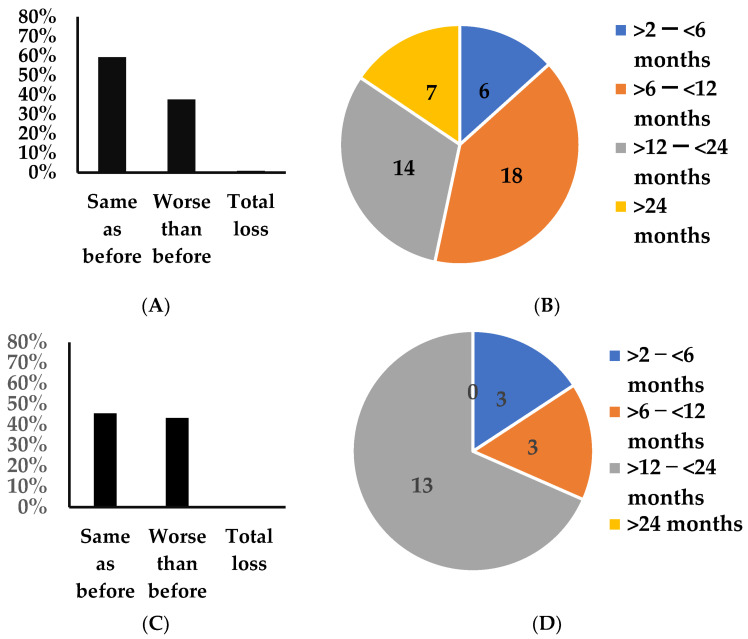
Characteristics of changes in smell perception. (**A**) Distribution of altered smell among individuals in the long COVID (**A**) and COVID reinfection (**C**) cohorts. Distribution of duration of taste perception worse than pre COVID in the long COVID (**B**) and COVID reinfection (**D**) cohorts.

**Figure 4 jcm-12-03598-f004:**
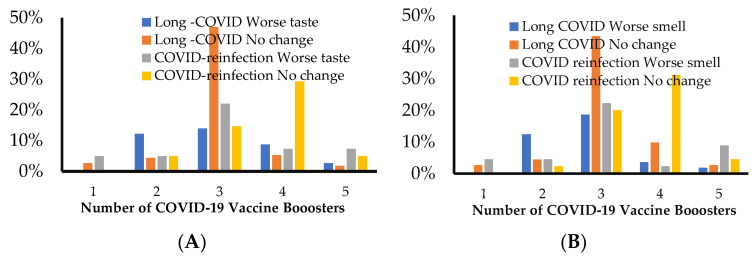
Correlation of vaccinations with taste and smell changes in long COVID and COVID reinfection. Shows distribution of number of individuals (as percent of the total cohort) experiencing altered taste (**A**) or smell (**B**) with respect to the number of COVID-19 vaccine boosters received.

**Figure 5 jcm-12-03598-f005:**
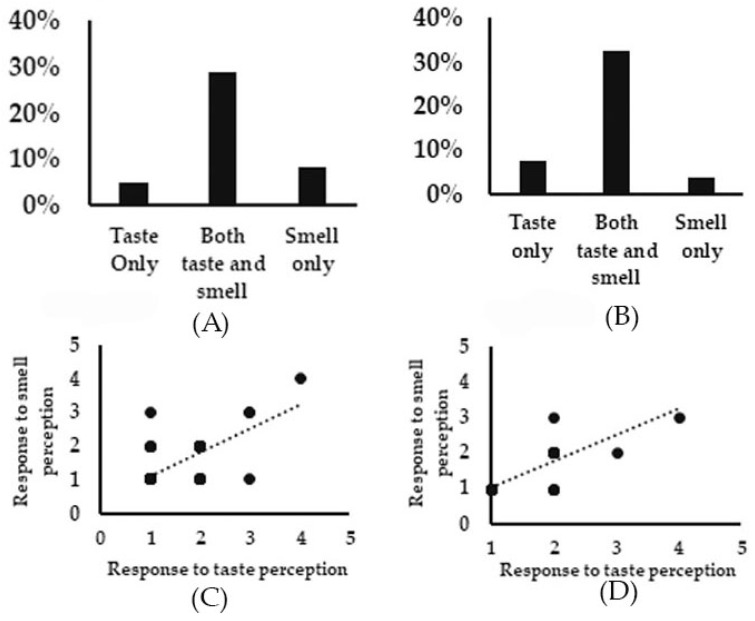
Correlation of taste and smell changes. (**A**) A higher percentage of individuals reported both taste and smell perception worse than pre COVID in both the long COVID (**A**) and COVID reinfection (**B**) cohorts. (**C**) Pearson correlation (r) between taste and smell worse than pre COVID in long COVID (**C**) and COVID reinfection (**D**) cohorts.

**Table 1 jcm-12-03598-t001:** (A): Participant characteristics of the long COVID and COVID reinfection cohorts. (B): Frequency distribution.

**(A)**			
	Long COVID (N = 127)	COVID reinfection (N = 47)	*p* value
Age	46.7 ± 17.8 yrs	45.8 ± 13.9 yrs	0.37
Sex	29 M:92 F *	11 M:36 F	
Mean duration	352.4 ± 250 days	488 ± 228.5 days	0.0007
Vaccination: % N (V + booster)	2% (1 + 0)	6% (1 + 0)	0.54
	19% (1 + 1)	15% (1 + 1)	
	57% (1 + 2)	55% (1 + 2)	
	13% (1 + 3)	13% (1 + 3)	
	6% (1 + 4)	11% (1 + 4)	
	2% (1 + 5)	0% (1 + 5)	
**(B)**			
		Number of individuals	
Duration (days)		Long COVID	COVID reinfection
60–119		25	4
120–179		9	1
180–239		31	5
240–299		13	6
300–359		7	1
360–419		2	0
420–479		0	0
480–539		0	4
540–599		8	4
600–659		9	8
660–719		10	9
720–779		4	3
780–839		2	0
840–899		6	2
900–959		1	4
Total		127	47
History of hospitalization		6	4

* no response was provided in six surveys.

**Table 2 jcm-12-03598-t002:** (A) Association between duration and taste perception in long COVID and COVID reinfection. (B) Association between duration and smell perception in long COVID and COVID reinfection.

**(A)**				
Taste	Long COVID		COVID Reinfection	
Duration	No Changes (82)	Worse ^@^ (35)	No changes (23)	Worse ^@^ (18)
2 to 6 months	27 (23%)	4 (3%)	3 (7%)	1 (2%)
>6 to 12 months	36 (31%)	13 (11%)	8 (20%)	4 (10%)
>12 to 24 months	16 (14%)	11 (9%)	9 (22%)	12 (29%)
>24 months	3 (3%)	7 (6%)	3 (7%)	1 (2%)
Chi-square	13.72		3.2	
*p* value	0.003		0.36	
**(B)**				
Smell	Long COVID		COVID reinfection	
	No changes (71)	Worse ^@^ (45)	No changes (22)	Worse ^@^ (19)
2 to 6 months	25 (22%)	6 (5%)	4 (7%)	3 (7%)
>6 to 12 months	30 (26%)	18 (16%)	3 (10%)	3 (7%)
>12 to 24 months	13 (11%)	14 (12%)	11 (27%)	13 (32%)
>24 months	3 (3%)	7 (6%)	4 (10%)	0
Chi-square	11.1		4.1	
*p* value	0.012		0.25	

^@^ = worse than pre COVID. The absolute number for the indicated response is given with the percentage with respect to the total number in the cohort in parenthesis.

**Table 3 jcm-12-03598-t003:** Comparison of dysgeusia and dysosmia across the specified duration between long COVID and COVID reinfection cohorts.

Worse than Pre COVID		<12 Months	12–24 Months	>24 Months	Chi-Square	*p*-Value
Taste	Long COVID (35)	17 (49%)	11 (31%)	7 (20%)	6.23	0.043
	COVID reinfection (18)	5 (28%)	12 (67%)	1 (6%)		
Smell	Long COVID (45)	24 (53%)	14 (31%)	7 (16%)	17.56	0.0002
	COVID reinfection (19)	6 (32%)	13 (68%)	0		

**Table 4 jcm-12-03598-t004:** Association between long COVID or COVID reinfection with headache or joint pain.

		Change from Pre COVID				
					Chi-Square	*p*-Value
Long COVID	Headache	78 (69%) (Never and rarely)	27 (20%) Sometimes	17 (10%) Often and always	17.16	0.0002
	Taste	81 (66%) (No change)	41 (33%) Worse	1 (1%) Total loss		
COVID reinfection	Headache	23 (50%)	18 (39%)	11 (24%)	9.25	0.0098
	Taste	27 (59%)	23 (50%)	1 (2%)		
Long COVID	Joint pain	82 (65%)	20 (15%)	24(19%)	28.36	0.0000007
	Taste	81 (66%)	41 (33%)	1 (1%)		
COVID reinfection	Joint pain	29 (63%)	10 (22%)	13 (28%)	15.5	0.00044
	Taste	27 (59%)	23 (50%)	1 (2%)		
Long COVID	Headache	78 (69%)	27 (20%)	17 (10%)	18.71	0.000087
	Smell	78 (63%)	45 (34%)	1 (1%)		
COVID reinfection	Headache	23 (50%)	18 (39%)	11 (24%)	10.04	0.0066
	Smell	25 (54%)	19 (41%)	0		
Long COVID	Joint pain	82 (65%)	20 (15%)	24 (19%)	30.86	0.0000002
	Smell	78 (63%)	45 (34%)	1 (1%)		
COVID reinfection	Joint pain	29 (63%)	10 (22%)	13 (28%)	15.5	0.00044
	Smell	25 (54%)	19 (41%)	0		

## Data Availability

The data presented in this study are available on request from the corresponding author. The data are not publicly available due to individual privacy and ethical reasons.

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
