# Peer review of "Characteristics of Chemosensory Perception in Long COVID and COVID Reinfection"

_jcm, 2023, doi:10.3390/jcm12103598_

Round 1
Reviewer 1 Report
The topic considered is interesting and the cases collected are relevant. In the introductory part, I would suggest to add (line 36-40) some references on the difficulty of diagnosing allergic pathologies such as rhinitis which can often cause overlapping symptoms including anosmia. ( see: Brindisi G, De Vittori V, De Nola R, Pignataro E, Anania C, De Castro G, Cinicola B, Gori A, Cicinelli E, Zicari AM. Updates on Children with Allergic Rhinitis and Asthma during the COVID-19 Outbreak. J Clin Med. 2021 May 24;10(11):2278. doi: 10.3390/jcm10112278. PMID: 34073986; PMCID: PMC8197398.
Brindisi G, De Vittori V, De Castro G, Duse M, Zicari AM. Pills to think about in allergic rhinitis children during COVID-19 era. Acta Paediatr. 2020 Oct;109(10):2149-2150. doi: 10.1111/apa.15462. Epub 2020 Jul 19. PMID: 32627237; PMCID: PMC7361544.
Brindisi G, Spalice A, Anania C, Bonci F, Gori A, Capponi M, Cinicola B, De Castro G, Martinelli I, Pulvirenti F, Matera L, Mancino E, Guido CA, Zicari AM. COVID-19, Anosmia, and Allergies: Is There a Relationship? A Pediatric Perspective. J Clin Med. 2022 Aug 26;11(17):5019. doi: 10.3390/jcm11175019. PMID: 36078947; PMCID: PMC9457095.)
In addition I would suggest to add (line 34) also the concept of Long Covid in pediatric age in which it was more relevant of the acute infection itself(Parisi GF, Diaferio L, Brindisi G, Indolfi C, Umano GR, Klain A, Marchese G, Ghiglioni DG, Zicari AM, Marseglia GL, Miraglia Del Giudice M. Cross-Sectional Survey on Long Term Sequelae of Pediatric COVID-19 among Italian Pediatricians. Children (Basel). 2021 Aug 31;8(9):769. doi: 10.3390/children8090769. PMID: 34572201; PMCID: PMC8467017.)
Good description of the results and of the statistical analysis of data. Minor english correction are requested
Minor English corrections of the forms of few sentences.
Author Response
Point by point Response:
Reviewer 1:
The topic considered is interesting and the cases collected are relevant. Good description of the results and of the statistical analysis of data.
Response: We thank the reviewer for the encouraging comment.
Critique #1: In the introductory part, I would suggest to add (line 36-40) some references on the difficulty of diagnosing allergic pathologies such as rhinitis which can often cause overlapping symptoms including anosmia.
Critique #2: In addition I would suggest to add (line 34) also the concept of Long Covid in pediatric age in which it was more relevant of the acute infection itself.
Response: We thank the reviewer for the suggestions. The critiques were combined for ease of review. Please review the following changes incorporated in the revised manuscript.
- Brindisi G, Spalice A, Anania C, Bonci F, Gori A, Capponi M, Cinicola B, De Castro G, Martinelli I, Pulvirenti F et al: COVID-19, Anosmia, and Allergies: Is There a Relationship? A Pediatric Perspective. J Clin Med 2022, 11(17). Page 12: 387-389.
- Brindisi G, De Vittori V, De Nola R, Pignataro E, Anania C, De Castro G, Cinicola B, Gori A, Cicinelli E, Zicari AM: Updates on Children with Allergic Rhinitis and Asthma during the COVID-19 Outbreak. J Clin Med 2021, 10(11). Page 12: 394-396.
Critique #3: Minor english correction are requested.
Response: We edited the revised manuscript thoroughly. Thank you.

Reviewer 2 Report
This paper indicates the frequency of long COVID symptoms in a cohort of self-reported patients that experienced SARS-CoV-2 infection at least once. The paper requires a moderate revision in terms of grammar, and a deep revision of the contents.
Page 1 lines 10-11 “affected with the coronavirus disease-19 (COVID-19)”. “With” is not the correct preposition.
Page 1 line 19 “Worse taste perception than pre-COVID”. Please check if the words are in the correct order.
Page 1 lines 20-21 “Smell perception worse than 20 pre-COVID were reported by 12% and 29% of individuals in the long-COVID and COVID-reinfection cohorts, respectively.” Please check the subject-verb agreement.
Page 1 line 30 “severe acute respiratory syndrome coronavirus-2 30 (CoV2)”. The acronym is indicated as CoV2 here and at line 35, and as SARS-CoV2 at line 34. Please check the official nomenclature and correct the wrong acronyms.
Page 1 lines 35-37 “the condition is defined as symptoms that occur in individuals with history of probable or confirmed CoV2 infection that begin within three months of the onset of COVID and lasts at least 2 months and cannot be explained by an alternate diagnosis”. Please check the subject-verb agreement and try to make the syntax simpler.
Page 2 lines 48-50 “In this study, we report a detailed analysis of population-based, self-reported survey data from hospitalized and non-hospitalized individuals with history of 49 SARS-CoV2 positive test.” The method for the collection of data left this reviewer perplexed. How did the authors verified if the subjects were/had been true COVID positive subjects? Did the authors discriminate between hospitalized and non-hospitalized individuals only on the basis of self-reported pieces of information? Please clearly state this in the text, since it may represent a strong limitation. Also, the authors put some emphasis on the fact that their data include details about both hospitalized and non-hospitalized individuals. However, no specific comparison was performed between these two groups. If the condition hospitalization vs non-hospitalization is not useful in terms of specific (i.e., centred on hospitalization) analysis, the concept should be eliminated from this sentence and from page 7 line 221.
Page 2 lines 69-70 “For positive COVID tests, the responders reported the date of initial positive testing and the number of times they tested positive subsequently”. As the authors wrote at page 1 lines 36-37, the symptoms indicated as Long COVID “begin within three months of the onset of COVID and lasts at least 2 months and cannot be explained by an alternate diagnosis”. However, in the COVID reinfection cohort it is not clear if the condition began after the first infection and lasted during/after the following re-infections, or if long COVID appeared as a result of the second/third/and so on infections. Please kindly add this piece of data in the tables.
Page 2 lines 71-73 “Response to symptom specific items were recorded on a five-point or four-point scale (all the time, most of the time, some of the time, a little of the time and none of the time).” Is there any sort of quantification? For example, some of the time = twice a week? Was everything left to the interpretation of the responders? Also, why the scale in Figure 1 (Page 4) is different? Please explain in the text and adopt the same scale in the text and the figure.
Page 2 liner 88-89 “Of the 225 responders, 57 were males and 157 were 88 females and the rest chose not to identify themselves”. In the table the only answers indicated are male and female. Were all the other subjects excluded from the analysis?
Page 3 lines 94-122 This section is pretty confusing. The authors should begin their dissertation reporting clearly the criteria that they used to separate the subjects into two cohorts (long-COVID and COVID-reinfection). Surely the authors may make some comments about the cases that were excluded from the analysis, however they should include a detailed description of the data only for those subjects that were actually analysed in this study. Instead the text is not clear in multiple parts. For example, the authors wrote (lines 114-116) “The duration since the first positive COVID test ranged between 21 and 875 days with an average of 467 days or 15 months and median of 556 days.” This piece of information belongs to a starting group of 53 subjects, and is reported in Table 1A, however Table 1A contains the detail of the definitive COVID-reinfection cohort, made of 46 subjects. It is not understandable to which group (starting vs definitive) the table refers. Also, at line 114 the authors mention a “Table 1”, that does not exist. Moreover, this section is misleading. For example, (lines 94-98) “Amongst the 225 responders, 76% (172) reported testing positive for COVID one time and 24% (53) more than once. The average duration since testing positive for the cohort with one positive COVID test is 275 days (Table 1A). Over 30% experienced at least one symptom persistent long COVID, this group is categorized as long-COVID cohort”. If not all these 172 subjects experienced long-COVID, then these subjects should not be indicated as the long-COVID group. Please kindly rewrite this whole paragraph following a logical order. Also, please rename the long-COVID cohort (whose only 30% of members actually experienced long-COVID) with another name.
Page 4 lines 137-141 “The frequency of symptoms that was experienced sometimes was higher for chest pain (37%) followed by headache (33%), cough (20%), 138 joint pain (19%) and muscle pain (17%) (Fig 1 B). In this cohort, headache, joint pain, cough, and muscle pain was reported to be present often and always by 8%, 11%, 22% and 15% for two years.” Please check the subject-verb agreement.
Page 4 lines 151-152 “We analyzed the relative frequency of taste dysfunction only in individuals experiencing the symptom for longer than three months.” As the authors wrote at page 1 lines 36-37, the symptoms indicated as long-COVID “begin within three months of the onset of COVID and lasts at least 2 months and cannot be explained by an alternate diagnosis”. So why did the authors choose to exclude patients experiencing long-COVID for 2 up to 3 months? Please explain in the text.
Page 5 line 178 “In the COVID reinfection cohort, 40% (19/47)”. The cohort is made of 46 subjects, not 47.
Page 6 lines 187-190 “In both long-COVID and COVID-reinfection cohorts, greater number of individuals reported experiencing both taste and smell perception worse than pre-COVID than either taste or smell dysfunction alone as is evident by the r value of 0.73 and 0.8 respectively (Fig 4A, B, C)”. The r value in the figure is 0.7 in all the diagrams.
Page 8 lines 231-232 “In contrast, COVID-19 patients reported loss of taste or smell without nasal congestion or discharge.”. By now, is SARS-CoV-2 the only identified pathogen causing a loss of taste/smell without nasal congestion or discharge? Please discuss this in the text. Also, add more recent references.
Page 8 lines 250-251 “This does not seem to be related to the frequency of infection or the number of boosters received.” Did the authors perform a statistical analysis to demonstrate this aspect? Please, explain in the text.
Page 8 line 269 “our data is consistent”. Please check the subject-verb agreement.
Page 8 line 271 “Repeat infections is”. Please check the subject-verb agreement.
Page 8 line 272-273 “Since chemosensory dysfunction have been strongly associated with neurological pathologies”. Please check the subject-verb agreement.
A modest language check is necessary.
Author Response
Point by point Response:
Reviewer 2:
This paper indicates the frequency of long COVID symptoms in a cohort of self-reported patients that experienced SARS-CoV-2 infection at least once. The paper requires a moderate revision in terms of grammar, and a deep revision of the contents.
Critique #1: Page 1 lines 10-11 “affected with the coronavirus disease-19 (COVID-19)”. “With” is not the correct preposition.
Response: Changed as “affected by the COVID.
Emerging data suggest an increasing prevalence of persistent symptoms in individuals affected by coronavirus disease-19 (COVID-19). Page 1: Lines 10-11.
Critique #2: Page 1 line 19 “Worse taste perception than pre-COVID”. Please check if the words are in the correct order.
Response: Changed, thank you.
Taste perception worse than pre-COVID was reported by 29% and 42% of individuals in the long-COVID and COVID-reinfection cohorts respectively. Page 1: lines 19-20.
Critique #3: Page 1 lines 20-21 “Smell perception worse than pre-COVID were reported by 12% and 29% of individuals in the long-COVID and COVID-reinfection cohorts, respectively.” Please check the subject-verb agreement.
Response: Done, thank you. The following text is included in the revised manuscript.
Smell perception worse than pre-COVID was reported by 37% and 46% of individuals in the long-COVID and COVID-reinfection cohorts respectively. Page 1: lines 20-22.
Critique #4: Page 1 line 30 “severe acute respiratory syndrome coronavirus-2 30 (CoV2)”. The acronym is indicated as CoV2 here and at line 35, and as SARS-CoV2 at line 34. Please check the official nomenclature and correct the wrong acronyms.
Response: Changed the abbreviation to CoV-2.
Critique #5: Page 1 lines 35-37 “the condition is defined as symptoms that occur in individuals with history of probable or confirmed CoV2 infection that begin within three months of the onset of COVID and lasts at least 2 months and cannot be explained by an alternate diagnosis”. Please check the subject-verb agreement and try to make the syntax simpler.
Response: Thank you, corrected.
Post-acute sequelae of SARS-CoV-2 infections (PASC) or Long COVID, the condition is defined as symptoms that occur in individuals with history of probable or confirmed CoV-2 infection that begins within three months of the onset of COVID and lasts at least 2 months and cannot be explained by an alternate diagnosis [5]. Page1, lines 34-37.
Critique #6: Page 2 lines 48-50 “In this study, we report a detailed analysis of population-based, self-reported survey data from hospitalized and non-hospitalized individuals with history of SARS-CoV2 positive test.” The method for the collection of data left this reviewer perplexed. How did the authors verified if the subjects were/had been true COVID positive subjects?
Response: We thank the reviewer for the critical comment. The survey responders self-reported COVID positive test and the frequency of testing positive. Further, since all respondents are volunteers from the Indiana University Health hospital COVID-registry, the self-reporting is more likely consistent with the laboratory test.
In response to the reviewer’s comment, the following changes are incorporated in the revised text.
The survey was distributed to 13,561 volunteers. Two hundred and twenty-five responders self-reported the date and the number of times they had COVID-19 positive test results. Page 2: lines 93-94.
Critique #7: Did the authors discriminate between hospitalized and non-hospitalized individuals only on the basis of self-reported pieces of information? Please clearly state this in the text since it may represent a strong limitation. Also, the authors put some emphasis on the fact that their data include details about both hospitalized and non-hospitalized individuals. However, no specific comparison was performed between these two groups. If the condition hospitalization vs non-hospitalization is not useful in terms of specific (i.e., centred on hospitalization) analysis, the concept should be eliminated from this sentence and from page 7 line 221).
Response: We agree with the reviewer that the persistence and/or recovery from long-COVID symptoms have been associated with COVID related hospitalization or ambulatory patients. In our cohort of long-COVID only 8 reported history of hospitalization (0.5%) and in the COVID-reinfection cohort 4 reported hospitalization (8%). With respect to the reviewer’s comment, we excluded data from individuals with history of hospitalization in the analysis and report the revised data in the re-submission. The following changes are incorporated in the revised submission.
One hundred and seventy-two individuals reported single COVID-19 positive test. The duration since testing positive ranged between 6 and 906 days with a median of 222 days (Table 1A). Only data from individuals testing positive at least 60 days prior to the date of responding to the survey was selected for analysis (Table 1B). This included a total of 127 respondents and constituted the long-COVID cohort. There were more females (92) than males (29) in this cohort. This is consistent with the reports of higher preponderance of females being diagnosed with long-COVID [21, 22]. Eight individuals reported hospitalization due to COVID in this cohort. Amongst these individuals, the duration of persistent symptoms was less than two months in two individuals, one individual each experienced taste perception worse than pre-COVID for 4 months and 28 months respectively and four did not experience any change post-COVID. We excluded individuals with history of hospitalization due to COVID in further analysis to minimize confounding factors. Page 3: lines 98-110.
Fifty-two individuals self-reported testing positive for COVID-19 two times or more. Of these, 78.8% (41) tested positive twice, 17.3% (9) tested positive three times and two individuals (3.8%) reported four COVID positive tests (Table 1). The duration since the first positive COVID test ranged between 21 and 875 days with a median of 556 days. Responses from 5 individuals with two positive COVID tests and duration of symptoms less than 60 days were not included for further analysis. The remaining cohort with 47 individuals (11 males and 36 females) with the minimum duration of 101 days or three months constituted the COVID reinfection cohort. Four individuals reported history of hospitalization due to COVID in this cohort and data from these individuals were excluded in further analysis to minimize confounding factors. Page 3: lines 111-120.
Critique #8: Page 2 lines 69-70 “For positive COVID tests, the responders reported the date of initial positive testing and the number of times they tested positive subsequently”. As the authors wrote at page 1 lines 36-37, the symptoms indicated as Long COVID “begin within three months of the onset of COVID and lasts at least 2 months and cannot be explained by an alternate diagnosis”. However, in the COVID reinfection cohort it is not clear if the condition began after the first infection and lasted during/after the following re-infections, or if long COVID appeared as a result of the second/third/and so on infections. Please kindly add this piece of data in the tables.
Response: We appreciate the excellent comment by the reviewer. In response we include the following description/explanation in the revised text.
However, it is relevant to note here that since all individuals were responding to a survey question on the experience of these symptoms post-positive COVID-19 test, it is not known as to whether they are reporting symptoms being experienced after the first infection that persisted and/or increased or symptoms that began after subsequent re-infections. Page 4: lines 151-155.
Our data does not specify the period between COVID-19 positive tests, and hence could potentially include repeat positive tests in relation to first infection in the re-infection co-hort. However, since we analyzed only data from responders reporting symptoms that lasted longer than two months and >60% of individuals experienced symptoms for longer than one year and 10% for longer than two years, it is likely that the responses represent symptoms that persist after first positive COVID-19 test in the COVID re-infection cohort also. Page 10: lines 316-322.
Critique #9: Page 2 lines 71-73 “Response to symptom specific items were recorded on a five-point or four-point scale (all the time, most of the time, some of the time, a little of the time and none of the time).” Is there any sort of quantification? For example, some of the time = twice a week? Was everything left to the interpretation of the responders? Also, why the scale in Figure 1 (Page 4) is different? Please explain in the text and adopt the same scale in the text and the figure.
Response: We apologize for the tardiness in presentation. Consistent with many electronic surveys, we used a qualitative scale to record responses. The text is revised in the resubmission.
Response to symptom specific items were recorded on a five-point or four-point scale (all the time, most of the time, some of the time, a little of the time and none of the time). Response to taste and smell specific questions were recorded on a four-point scale (same as before, worse than before, better than before and total loss). Page 2: lines 75-80.
Critique #10: Page 2 liner 88-89 “Of the 225 responders, 57 were males and 157 were 88 females and the rest chose not to identify themselves”. In the table the only answers indicated are male and female. Were all the other subjects excluded from the analysis?
Response: The options for gender question in our survey included male, female and choose not to identify. Some chose the third option. This is indicated in the revised table 1A. Page 3: lines 121-128.
Table 1 A: Participant characteristics of the long-COVID and COVID-reinfection cohorts.
* no response was provided in six surveys.
Critique #11: Page 3 lines 94-122 This section is pretty confusing. The authors should begin their dissertation reporting clearly the criteria that they used to separate the subjects into two cohorts (long-COVID and COVID-reinfection). Surely the authors may make some comments about the cases that were excluded from the analysis, however they should include a detailed description of the data only for those subjects that were actually analysed in this study. Instead the text is not clear in multiple parts. For example, the authors wrote (lines 114-116) “The duration since the first positive COVID test ranged between 21 and 875 days with an average of 467 days or 15 months and median of 556 days.” This piece of information belongs to a starting group of 53 subjects, and is reported in Table 1A, however Table 1A contains the detail of the definitive COVID-reinfection cohort, made of 46 subjects. It is not understandable to which group (starting vs definitive) the table refers. Also, at line 114 the authors mention a “Table 1”, that does not exist. Moreover, this section is misleading. For example, (lines 94-98) “Amongst the 225 responders, 76% (172) reported testing positive for COVID one time and 24% (53) more than once. The average duration since testing positive for the cohort with one positive COVID test is 275 days (Table 1A). Over 30% experienced at least one symptom persistent long COVID, this group is categorized as long-COVID cohort”. If not all these 172 subjects experienced long-COVID, then these subjects should not be indicated as the long-COVID group. Please kindly rewrite this whole paragraph following a logical order. Also, please rename the long-COVID cohort (whose only 30% of members actually experienced long-COVID) with another name.
Response: We apologize for the lack of clarity. Please review the revised text below, incorporated in the revised submission.
The survey was distributed to 13,561 volunteers. Two hundred and twenty-five responders self-reported the date and the number of times they had COVID positive test results. Of the 225 responders, 57 were males and 157 were females and the rest chose not to identify themselves. The mean age was 45.8 years (range: 19-84 years). One hundred and ninety-four identified as white, 18 as African American, 7 as Asian, 3 as Hispanic and rest did not choose to report their race. Page 2: lines 93-97.
One hundred and seventy-two individuals reported single COVID-19 positive test. The duration since testing positive ranged between 6 and 906 days with a median of 222 days (Table 1A). Only data from individuals testing positive at least 60 days prior to the date of responding to the survey was selected for analysis (Table 1B). This included a total of 127 respondents and constituted the long-COVID cohort. There were more females (92) than males (29) in this cohort. This supports the previous reports of higher preponderance of females diagnosed with long-COVID [21, 22]. Eight individuals reported hospitalization due to COVID in this cohort. Amongst these individuals with history of hospitalization, the duration of persistent symptoms was less than two months in two individuals, one individual each experienced taste perception worse than pre-COVID for 4 months and 28 months respectively and four did not experience any change post-COVID. We excluded individuals with history of hospitalization due to COVID to minimize confounding factors. Page 3: lines 98-110.
Fifty-two individuals self-reported testing positive for COVID-19 two times or more. Of these, 78.8% (41) tested positive twice, 17.3% (9) tested positive three times and two individuals (3.8%) reported four COVID-19 positive tests (Table 1A). The duration since the first positive COVID-19 test ranged between 21 and 875 days with a median of 556 days. Responses from 5 individuals with two positive COVID-19 tests and duration of symptoms less than 60 days were not included for further analysis. The remaining cohort with 47 individuals (11 males and 36 females) with the minimum duration of 101 days or three months constituted the COVID reinfection cohort (Table 1B). Four individuals reported history of hospitalization due to COVID in this cohort and data from these individuals were excluded in further analysis to minimize confounding factors. Page 3: lines 111-120.
All individuals in this study were vaccinated, with 57% in the long-COVID cohort and 55% in the COVID-reinfection cohort receiving two boosters. In the COVID-reinfection cohort, 10% received four boosters and in the long-COVID cohort 2% received five boosters (Table 1) Page 4: lines 123-126.
Critique #12: Page 4 lines 137-141 “The frequency of symptoms that was experienced sometimes was higher for chest pain (37%) followed by headache (33%), cough (20%), joint pain (19%) and muscle pain (17%) (Fig 1 B). In this cohort, headache, joint pain, cough, and muscle pain was reported to be present often and always by 8%, 11%, 22% and 15% for two years.” Please check the subject-verb agreement.
Response: Corrected, thank you. Please review the revised text given below.
The frequency of symptoms that was experienced sometimes was higher for chest pain and headache (35%) followed by cough (20%), joint pain (19%) and muscle pain (19%) (Fig 1 B). In this cohort, headache, joint pain, cough, and muscle pain were reported to be present often and always by 8%, 11%, 22% and 15% for two years (data not shown) Page 4: lines 147-151.
Critique #13: Page 4 lines 151-152 “We analyzed the relative frequency of taste dysfunction only in individuals experiencing the symptom for longer than three months.” As the authors wrote at page 1 lines 36-37, the symptoms indicated as long-COVID “begin within three months of the onset of COVID and lasts at least 2 months and cannot be explained by an alternate diagnosis”. So why did the authors choose to exclude patients experiencing long-COVID for 2 up to 3 months? Please explain in the text.
Critique #14: Page 5 line 178 “In the COVID reinfection cohort, 40% (19/47)”. The cohort is made of 46 subjects, not 47.
Response: We agree with the reviewer’s comments. In the revised submission, data from two months were included in the calculations. This entailed adding responses from 12 individuals and one more person in the revised calculations of both the long-COVID and COVID-reinfection cohorts.
Please review the revised text- response to critique #11 above. Please review the revised figures and tables.
Page 5: lines 180-187.
Critique #15: Page 6 lines 187-190 “In both long-COVID and COVID-reinfection cohorts, greater number of individuals reported experiencing both taste and smell perception worse than pre-COVID than either taste or smell dysfunction alone as is evident by the r value of 0.73 and 0.8 respectively (Fig 5 A, B, C)”.
Response: Corrected in the text, thank you. Page 8: line 240.
Critique #16: Page 8 lines 231-232 “In contrast, COVID-19 patients reported loss of taste or smell without nasal congestion or discharge.”. By now, is SARS-CoV-2 the only identified pathogen causing a loss of taste/smell without nasal congestion or discharge? Please discuss this in the text. Also, add more recent references.
Response: The sentence is referring to the associated citation. To clarify, the sentence is changed as follows in the revised text.
In contrast, in a study comparing gustatory functions in patients affected by COVID-19 and or common cold, it was observed that the sweet and bitter taste scores were significantly worse in COVID-19 patients without nasal congestion or discharge [29]. Page 9: lines 281-284.
Critique #17: Page 8 lines 250-251 “This does not seem to be related to the frequency of infection or the number of boosters received.” Did the authors perform a statistical analysis to demonstrate this aspect? Please, explain in the text.
Response: With respect to the reviewer’s comment, the following text and new figures are incorporated in the revised text.
- Correlation between vaccination and smell/taste perception in long-COVID and COVID reinfection:
In both the long-COVID and the COVID-reinfection cohorts, a higher percentage of individuals that received two boosters experienced worse taste (14% and 22% respectively) or smell perception (19% and 22% respectively) (Fig 4 A, B). Interestingly, the number of individuals reporting chemosensory dysfunction decreased precipitously with increasing number of boosters with 4% and 2% experiencing altered smell in the long-COVID and COVID-reinfection respectively. However, this is attributed to the highest number of individuals (Table 1) receiving two boosters as opposed to 20% receiving three or four boosters in our study cohort. The association between the number of boosters and the du-ration of worse taste/smell perception was not significant with chi square value of 12.8 and 9 respectively, p< 0.3. Page 7, lines 221-233 and Page 8, lines 234-236.
Fig 4A Fig 4B
Figure 4: Correlation of vaccinations with taste and smell changes in long-COVID and COVID reinfection. Shows distribution of number of individuals (as percent of the total cohort) experiencing altered taste (A) or smell (B) with respect to the number of COVID-19 vaccine boosters received.
Critique #18: Page 8 line 269 “our data is consistent”. Please check the subject-verb agreement.
Response: Thank you, corrected.
In summary, our data are consistent with the observations that the long-COVID characterized by persistent multi-organ symptoms affects a proportion of COVID infected individuals despite vaccination. Page 10: 334-336.
Critique #19: Page 8 line 271 “Repeat infections is”. Please check the subject-verb agreement.
Response: Corrected, thank you.
Repeat infections are more likely associated with altered chemosensory perception for extended periods. Page 10: 336-337.
Critique #20: Page 8 line 272-273 “Since chemosensory dysfunction have been strongly associated with neurological pathologies”. Please check the subject-verb agreement.
Response: Corrected, thank you.
Since chemosensory dysfunction has been strongly associated with neurological pathologies, depression and inadequate quality of life, protective measures to prevent reinfection with SARS-CoV-2 is warranted. Page 10: lines 337-339.

Reviewer 3 Report
Jaramillo et al. represent the manuscript entitled: "Characteristics of chemosensory perception in long COVID 2 and COVID re-infection"
The manuscript is interesting, nicely written. The statistics used is appropriate. Discussion and conclusions are ok.
However, there are some points that need corrections:
line 29 where was this number about COVID-19 cases taken from?
line 57 COVID-19 instead of Covid-19 (please check in the whole text once again)
line 87 COVID-19 positive test result
line 95 COVID-19 one time
line 96 one positive COVID-19 test
line 148 long-COVID (check these small details in the whole text)
Table 1: some insertion errors
Table 2B: to be on one page if possible
line 298 extending the co ? (missing rest of the sentence)
Please go once again through the whole text and check if the abbreviations related to COVID-19, long-COVID etc. are correctly written.
Go once again through the whole text and check the typos. There are some is/are parts that need corrections.
Author Response
Reviewer 3:
Jaramillo et al. represent the manuscript entitled: "Characteristics of chemosensory perception in long COVID 2 and COVID re-infection"
The manuscript is interesting, nicely written. The statistics used is appropriate. Discussion and conclusions are ok.
Response: We thank the reviewer for the encouraging comments.
Critique #1: line 29 where was this number about COVID-19 cases taken from?
Response: We apologize for missing the reference. Please review the revised text.
Globally as of April 2023, there have been 762,201,169 confirmed cases of coronavirus disease 2019 (COVID-19) caused by the severe acute respiratory syndrome coronavirus-2 (CoV-2)[1]. Page 1: lines 29-31.
- WHO Coronavirus (COVID-19) Dashboard [https://covid19.who.int/WHO Page 11: line 370.
Critique # 2:
line 57 COVID-19 instead of Covid-19 (please check in the whole text once again)
line 87 COVID-19 positive test result
line 95 COVID-19 one time
line 96 one positive COVID-19 test
Response: The authors have combined the above critiques of the reviewer, so as to provide a
coherent response. We apologize for the tardiness; we corrected the same in the entire
manuscript. Please review the revised manuscript.
An electronic survey was sent to individuals aged 18 years and older, who had previously agreed to be notified about COVID-19 studies registered at IU School of Medicine’s COVID-19 Research Registry. Page 2: lines 59-61.
The survey was distributed to 13,561 volunteers and 225 responders self-reported the date and the number of times they had COVID-19 positive test results. Page 3, lines 93-94.
One hundred and seventy-two individuals reported single COVID-19 positive test. Page 3, line 98.
Fifty-two individuals self-reported testing positive for COVID-19 two times or more. Page 3, line 111.
Critique #3: line 148 long-COVID (check these small details in the whole text).
Response: We apologize for the tardiness; we corrected the same in the entire manuscript.
Critique #4: Table 1: some insertion errors.
Response: Please review the revised Table 1A. We also include a new Table 1B in the revised submission. Pge 3: 121-129, Page 4: line130
Table 1 A: Participant characteristics of the long-COVID and COVID-reinfection cohorts.
|
Long-COVID (N=127) |
COVID reinfection (N=47) |
p value |
|
|
Age |
46.7 +/- 17.8yrs |
45.8+/-13.9 yrs |
0.37 |
|
Sex |
29 M:92 F* |
11M:36F |
|
|
Mean duration |
352.4+/-250 days |
488+/-228.5 days |
0.0007 |
|
Vaccination: % N (V+booster) |
2% (1+0) 19% (1+1) 57% (1+2) 13% (1+3) 6% (1+4) 2% (1+5) |
6% (1+0) 15% (1+1) 55% (1+2) 13% (1+3) 11% (1+4) 0% (1+5) |
0.54 |
*no response was provided in six surveys
Table 1. B: Frequency distribution
|
|
Number of individuals |
|
|
Duration (days) |
Long -COVID |
COVID-reinfection |
|
60 - 119 |
25 |
4 |
|
120 - 179 |
9 |
1 |
|
180 - 239 |
31 |
5 |
|
240 - 299 |
13 |
6 |
|
300 - 359 |
7 |
1 |
|
360 - 419 |
2 |
0 |
|
420 - 479 |
0 |
0 |
|
480 - 539 |
0 |
4 |
|
540 - 599 |
8 |
4 |
|
600 - 659 |
9 |
8 |
|
660 - 719 |
10 |
9 |
|
720 - 779 |
4 |
3 |
|
780 - 839 |
2 |
0 |
|
840 - 899 |
6 |
2 |
|
900 - 959 |
1 |
4 |
|
Total |
127 |
47 |
|
History of hospitalization |
6 |
4 |
Critique # 5: Table 2B: to be on one page if possible.
Response: Done, please review the revised submission. Page 5, lines 180-188.
Critique #6: line 298 extending the co ? (missing rest of the sentence).
Response: Again, we apologize for the tardiness. Please review the revised submission.
We also thank Dr. Anubhuti Shukla for assisting in developing the survey and extending the study to oral health. Page 11: Lines 363-364.
Critique #7: Please go once again through the whole text and check if the abbreviations related to COVID-19, long-COVID etc. are correctly written.
Response: We edited the manuscript carefully. Thank you.

Reviewer 4 Report
Jaramillo and colleagues determined the relative frequency of altered taste and smell in COVID-reinfection and long-COVID cohort. Although this study presented interesting data, I stipulated below the comments that I would like the authors to address to further improve the quality of this paper. 1. It should be highlighted in the introduction that there are several factors that influence Long COVID development like age, sex, pre-existing comorbidities, and infecting SARS-CoV-2 variant. DOI: 10.1016/j.cmi.2021.11.002, DOI: 10.3390/jcm11247314; DOI: 10.3390/v14122629
2. For Table 1, kindly report the p-values per sociodemographic factor analyzed (e.g. age, sex, hospitalization, vaccination / booster status etc.)
3. Since the infecting SARS-CoV-2 variant could influence long COVID development, kindly report the dates the survey was administered as well as the circulating variant around that timeframe.
4. How did you ascertain that the indicated systemic symptoms reported by the long COVID cohort is truly attributable to long COVID and not by other pre-existing comorbidities that the cohort may have? The mean age of the cohort here is 45.8 years and some of whom might have succumbed to certain comorbidities that may share the same clinical presentation as long COVID.
5. Please indicate possible sources of bias in this study and the mitigations the researchers undertook to minimize bias.
Author Response
Reviewer 4.
Jaramillo and colleagues determined the relative frequency of altered taste and smell in COVID-reinfection and long-COVID cohort. Although this study presented interesting data, I stipulated below the comments that I would like the authors to address to further improve the quality of this paper.
Critique #1: It should be highlighted in the introduction that there are several factors that influence Long COVID development like age, sex, pre-existing comorbidities, and infecting SARS-CoV-2 variant. DOI: 10.1016/j.cmi.2021.11.002, DOI: 10.3390/jcm11247314; DOI: 10.3390/v14122629
Response: Thank you, please review the revised text.
Furthermore, age, sex, multiple pre-infection comorbidities (such as diabetes, asthma) and severity of acute CoV-2 infection (symptomatic/ asymptomatic, hospitalization) are confounding factors that could contribute to the development and/or persistence of heterogeneous post-COVID-19 conditions[2, 7]. Page 1: lines 40-44.
Critique #2: How did you ascertain that the indicated systemic symptoms reported by the long COVID cohort is truly attributable to long COVID and not by other pre-existing comorbidities that the cohort may have? The mean age of the cohort here is 45.8 years and some of whom might have succumbed to certain comorbidities that may share the same clinical presentation as long COVID.
Critique #3: Please indicate possible sources of bias in this study and the mitigations the researchers undertook to minimize bias.
Response: The authors have combined the above critiques of the reviewer, so as to provide a coherent response. We agree with the reviewer that the factors such as pre-existing comorbidities and the severity of initial infection could be confounders in data interpretation and is a limitation of the study. We are currently seeking approvals to access the electronic health records of these individuals to collect specific information on comorbidities, specific blood parameters and post-COVID hospitalization. Data will be disseminated subsequently.
With respect to the reviewer’s concern, the following changes have been incorporated in the revised manuscript. Please review.
Furthermore, age, sex, multiple pre-infection comorbidities (such as diabetes, asthma) and severity of acute CoV-2 infection (symptomatic/ asymptomatic, hospitalization) are confounding factors that could contribute to the development and/or persistence of heterogeneous post-COVID-19 conditions[2, 7]. Page 1: Lines 40-44
This included a total of 127 respondents and constituted the long-COVID cohort. There were more females (92) than males (29) in this cohort. This supports the previous reports of higher preponderance of females diagnosed with long-COVID[19, 20]. Page 3: lines 101-104.
The remaining cohort with 47 (11 males and 36 females) individuals with the minimum duration of 101 days or three months constituted the COVID reinfection cohort (Table 1B). Page 3: lines 116-118.
Available literature suggests that the severity of smell and taste alterations is reduced in CoV-2 infected individuals during periods of Omicron variant dominance[31, 32]. However, in our study cohort the experience of altered (worse than pre-COVID) taste and smell perception does not seem to be related to the frequency of infection or the number of boosters received. This could be attributed to the difference in the global prevalence of different variants and positive-COVID test data in our cohort not-restricted to one variant. Interestingly, Notarte et al., in a systematic review, et al show that there is low level of evidence to suggest that vaccination before SARS-CoV-2 infection could reduce the risk of developing subsequent long-COVID[33]. Page 10: lines 300-308.
While we report age, sex, and race distribution, we did not perform symptom analyses by subgroup. Further, while the relative prevalence of altered chemosensation in our study is concomitant with previous studies[8, 14, 15], comorbidities such as diabetes that existed pre-COVID or newly developed post-COVID could have potentially contributed to the altered chemosensation in both long-COVID and COVID reinfection cohorts[2, 20]. Page 10: 324-328.
Critique #5: For Table 1, kindly report the p-values per sociodemographic factor analyzed (e.g. age, sex, hospitalization, vaccination / booster status etc.)
Response: Please review the revised table 1A.
Table 1 A: Participant characteristics of the long-COVID and COVID-reinfection cohorts.
|
Long-COVID (N=127) |
COVID reinfection (N=47) |
p value |
|
|
Age |
46.7 +/- 17.8yrs |
45.8+/-13.9 yrs |
0.37 |
|
Sex |
29 M:92 F* |
11M:36F |
|
|
Mean duration |
352.4+/-250 days |
488+/-228.5 days |
0.0007 |
|
Vaccination: % N (V+booster) |
2% (1+0) 19% (1+1) 57% (1+2) 13% (1+3) 6% (1+4) 2% (1+5) |
6% (1+0) 15% (1+1) 55% (1+2) 13% (1+3) 11% (1+4) 0% (1+5) |
0.54 |
*no response was provided in six surveys
Critique #6: Since the infecting SARS-CoV-2 variant could influence long COVID development, kindly report the dates the survey was administered as well as the circulating variant around that timeframe.
Response: We agree with the reviewer. Our survey data were collected over a period that encompassed the time when the Omicron and Delta variants were identified as Variants of Concern. However, the data include detailed information on the positive COVID-19 tests, which could be a limitation. With respect to the reviewer’s comments, the following changes are incorporated in the revised manuscript.
Data from surveys completed between August and September 2022 were analyzed in this study. Page 2: lines 84-85.
Available literature suggests that the severity of smell and taste alterations is reduced in CoV-2 infected individuals during periods of Omicron variant dominance[31, 32]. However, in our study cohort the experience of altered (worse than pre-COVID) taste and smell perception does not seem to be related to the frequency of infection or the number of boosters received. This could be attributed to the difference in the global prevalence of different variants and positive-COVID test in our cohort not-restricted to one variant. Interestingly, Notarte et al., in a systematic review, et al show that there is low level of evidence to suggest that vaccination before SARS-CoV-2 infection could reduce the risk of developing subsequent long-COVID[33]. Page 10: lines 300-308.

Round 2
Reviewer 2 Report
Thank you for addressing this reviewer's perplexities. The paper is now suitable for publication.
Reviewer 4 Report
The paper has been revised extensively and is now acceptable for publication.